Noninvasive and safe cell viability assay for Euglena gracilis using natural food pigment

Yamashita Kyohei 1
Yamada Koji 2
Suzuki Kengo 2
Tokunaga Eiji eiji@rs.kagu.tus.ac.jp 1
1 Department of Physics, Faculty of Science, Tokyo University of Science , Tokyo , Japan
2 euglena Co., Ltd. , Tsurumi-ku, Yokohama-shi, Kanagawa , Japan
Haraguchi Tokuko
Electronic publication date: 2019 Apr 4
Publication date: 2019
Volume: 7
Electronic Location ID: e6636
Received 2018 Oct 7; Accepted 2019 Feb 18
Copyright: ©2019 Yamashita et al.
Copyright year: 2019
Copyright holder: Yamashita et al.
License: This is an open access article distributed under the terms of the Creative Commons Attribution License, which permits unrestricted use, distribution, reproduction and adaptation in any medium and for any purpose provided that it is properly attributed. For attribution, the original author(s), title, publication source (PeerJ) and either DOI or URL of the article must be cited.
License URL: https://creativecommons.org/licenses/by/4.0/

Keywords: Anthocyanin, Dye exclusion test, Absorbance spectral imaging microscopy, Euglena, Monascus, Noninvasive cell viability assay, Natural food pigment, Trypan blue, Methylene blue, Unicellular photosynthetic green microalgae

Funding: The authors received no funding for this work.

==============================
Noninvasive and safe cell viability assay is required in many fields such as regenerative medicine, genetic engineering, single-cell analysis, and microbial food culture. In this case, a safe and inexpensive method which is a small load on cells and the environment is preferable without requiring expensive and space-consuming equipment and a technician to operate. We examined eight typical natural food pigments to find Monascus pigment (MP) or anthocyanin pigment (AP) works as a good viability indicator of dye exclusion test (DET) for Euglena gracilis which is an edible photosynthetic green microalga. This is the first report using natural food pigments as cell viability assay. Euglena gracilis stained by MP or AP can be visually judged with a bright field microscope. This was spectrally confirmed by scan-free, non-invasive absorbance spectral imaging A(x, y, λ) microscopy of single live cells and principal component analysis (PCA). To confirm the ability of staining dead cells and examine the load on the cells, these two natural pigments were compared with trypan blue (TB) and methylene blue (MP), which are synthetic dyes conventionally used for DET. As a result, MP and AP had as good ability of staining dead cells treated with microwave as TB and MB and showed faster and more uniform staining for dead cells in benzalkonium chloride than them. The growth curve and the ratio of dead cells in the culture showed that the synthetic dyes inhibit the growth of E. gracilis, but the natural pigments do not. As the cell density increased, however, AP increased the ratio of stained cells, which was prevented by the addition of glucose. MP can stain dead cells in a shorter time than AP, while AP is more stable in color against long-term irradiation of intense light than MP. Due to the low toxicity of these pigments, viability of cells in culture can be monitored with them over a long period.

Introduction

Unicellular microalgae E. gracilis has been drawing attention to realize a sustainable society of recycling resource and energy. E. gracilis is a photosynthetic flagellate green microalga with a length of approximately 50 µm and a diameter of 8 to 12 µm inhabiting freshwater (Wolken, 1967). Depending on nutritional and environmental conditions, E. gracilis synthesizes paramylon, a β-1,3-glucan, which is used as an ingredient of functional food (Sugiyama et al., 2009; Nakashima et al., 2018) or produces wax ester suited for its conversion to biofuel (Inui et al., 1982). By photosynthesis, E. gracilis can grow in both autotrophic culture (Cramer-Myers medium (Cramer & Myers, 1952)) and heterotrophic culture (Koren-Hutner medium (Kitaoka, 1989)). In recent years, the success of mass-cultivation of E. gracilis has enabled the commercial supply of E. gracilis as an ingredient of functional foods, cosmetics, and biofuel (Suzuki, 2017). Therefore, the technology using E. gracilis, which enables clean, sustainable cost-effective production of food and fuel, is being developed in industrial and academic fields (Yamada et al., 2016b). For the development of these technologies, noninvasive and safe cell viability assay plays an important role as a basic technology.

For example, the production of nuclear mutants of E. gracilis requires a physically and chemically significant load on cells due to the robustness of their genome, so it is necessary to confirm whether the cells are viable or dead at screening (Yamada et al., 2016a). Moreover, when the alga is mass cultured for edible use, it is necessary to confirm the state of growth.

As conventional methods to distinguish between live and dead cells in culture, the followings are known.

(1) Colony formation assay: The number of live cells is evaluated by the number of colonies formed on an agar culture medium after an inoculation of diluted cell culture and following definite time of culture (Collins & Lyne, 1985).

(2) Dye exclusion test (DET): A cell stained with a dye such as trypan blue (TB) is judged as a dead cell (Bonora & Mares, 1982).

(3) Enzyme activity assay: enzymatic reaction of enzymes in living cells or enzymes leaking out of dead cells are used for viability assay (Kaja et al., 2017).

(4) Flow cytometry analysis: dead cells is labeled with a fluorescent dye (Hamalainen-Laanaya & Orloff, 2012) and detected by fluorescence flow cytometry (Yamada et al., 2016b; Iwata et al., 2017).

(5) Optical method: the dead or alive state of cells is diagnosed by deflection change of the probe light beam (Wu & Terada, 2005).

However, these methods have drawbacks such as requiring specialized techniques and equipment, damaging cells, and inability to perform in-situ measurement in the cultivation process (Smith & March 1 P, inShare, 2013; Wu & Terada, 2005). Therefore, in order to solve these problems, in this paper, we propose a cell viability assay using natural edible pigments as the dye exclusion test (DET) in the above-mentioned method (2).

Trypan blue (TB) and methylene blue (MB), vital dyes, have been conventionally used for the DET. TB is a widely used diazo dye for selectively coloring dead tissues or cells. The mechanism for TB to stain cells is based on its negative charge which prevent the incorporation of it into the living cells with the membrane negatively charged. Therefore, the living cells are not stained, but the dead cells with the compromised cell membrane are stained by TB (ScienceDirect, 2018; Tran et al., 2011) environmental and cell health problems due to its potential teratogenic effects (Tsaousis et al., 2012; Beck & Lloyd, 1964). It is also pointed out that pore formation is possibly induced in cell membranes to increase membrane permeability (Tran et al., 2011). Methylene blue is frequently used to distinguish dead yeast cells from living cells (Tampion & Tampion, 1987). However, the DET method with methylene blue have suffered from false positive results at longer exposure times (Feizi et al., 2016). As other dyes for the DET, eosin (Schrek, 1936), nile blue (Scharff & Maupin, 1960), and amethyst violet (Novelli, 1962) have been used but it is known that the selective permeability of the plasma membrane is destroyed or severely impaired (Bonora & Mares, 1982), indicating that these dyes are toxic for cells.

In order to avoid these problems, a technique using erythrosin B (EB, aka Red No. 3) used as a food additive was developed (Kim et al., 2016). This synthetic colorant is a food dye that does not pass through biological membranes and is compatible with automatic cell counters. However, since EB has a property of fading in an acidic solution (Umezawa et al., 1990), it is not suitable for the culture of E. gracilis, which is ordinary culture at pH 3.5. In addition, since EB is suspected of being carcinogenic, once the FAD (Food and Drug Administration) banned their use (1990) (Associated Press, 1990; Jennings et al., 1990). Although the ban on EB was removed afterwards, most of the EB in the US has been replaced with Allura Red (aka Red 40) (Bell, 2017). Furthermore, in recent years, consumers are more conscious of ingredients in foods, and foods are required to be as “(facing the wrong way) natural” as possible (Downham & Collins, 0000; Giusti & Wrolstad, 2003), thus research on edible pigment extraction methods and food applications of them have been conducted (Akogou et al., 2018). Presently, E. gracilis is provided as a raw material for functional food (Nakashima et al., 2017). Therefore, if substitution with natural food pigments for DET is realized, the pigment is expected not only to reduce the load on the cells in viability assay in basic researches but also has potential to be used industrially as additive to the cultivation medium. Specifically, when natural food pigments are used for the viability assay in E. gracilis culture, there is no problem in using the harvested cells for food material even if they remain in the cell. Rather, residual pigments may enhance the value of product by adding antioxidant function possessed by natural pigments and increasing color variation as food material. These are advantages not found in synthetic food dyes.

Eight natural food pigments are tested in this study, among which Monascus pigment (MP) and anthocyanin pigment (AP) are of particular interest. MP is derived from Monascus sp. which is a kind of filamentous fungus and AP is derived from purple sweet potato. The colors (red or red purple) of these natural pigments correspond to the complementary color of vivid green of E. gracilis, so there is an advantage that the contrast becomes higher compared with other colors when observed by bright field microscopy.

MP exhibits a red color due to a molecule of the pigment whose main component is monascorubramine (Chen et al., 2017). It does not have a significant pH dependence of the color (although it tends to precipitate in acidic solution), is relatively stable to heat, and has excellent stainability to proteins (Shimizu, Nakamura & Fuji, 2001), while it is unstable against light irradiation, especially in an acidic condition. MP has been used for more than 1,000 years as a food pigment and a folk medicine in China because an efficient production method is established by fermentation of rice. MP is an inexpensive and reproducible substrate, has variation in colors, is highly safe, and shows good solubility in water and ethanol (Wang & Lin, 2007; Chen et al., 2017). Moreover, it possesses biological activities such as anti-mutagenic, anti-cancer properties, antibacterial activity, potential anti-obesity activity (Feng, Shao & Chen, 2012).

AP is a water-soluble pigment, dissolved as a glycoside in the vacuolar liquid of plants, and has a number of chemical structures depending on the type of sugar as well as the organic acid bound thereto (Shimizu, Nakamura & Fuji, 2001). AP turns red in acidic conditions and turns blue when pH increases. Many APs have high stability under acidic conditions compared to basic conditions (Khoo et al., 2017). In particular, the AP contained in purple sweet potatoes is superior in stability in heat and light resistance as compared with AP contained in other plants (Shimizu, Nakamura & Fuji, 2001). Also, AP interacts with both cellulose and pectin (Padayachee et al., 2012). AP has traditionally been used as natural food pigment. Furthermore, AP has a high antioxidant effect, and has preventive effects such as anti-diabetes drugs, anticancer drugs, anti-inflammatory drugs, anti-bacterial drugs, anti-obesity drugs, and anti-cardiovascular diseases (CVD) drugs (Khoo et al., 2017).

In this research, we use E. gracilis for the screen of food pigments which can be used for DET.

Materials and Methods

Sample preparation

Sample preparation for each experiment was conducted as follows.

The wild type E. gracilis (Z strain) was cultured in CM medium (Cramer-Myers medium (Cramer & Myers, 1952), pH 3.5), stationary and aerobically under continuous illumination with a cool white fluorescent light at 40 µmol/m2/sec and at constant temperature 21 °C before pigment was mixed. The cells were provided by euglena Co., Ltd. This suspension was injected into a micro test tube containing pigment and thoroughly mixed by pipetting. These samples were kept stationary in the micro test tubes with the cap opened under continuous illumination with a cool white fluorescent light at 100 µmol/m2/sec and at constant temperature 21 °C.

Observation of cells by bright field microscope

The bright field microscopes and objective lenses used for the observation of cells in this manuscript are as follows.

• Measurement of absorbance of single live or dead cells (Figs. 1–5). Inverted microscope (IX71, OLYMPUS) with the 100 × objective lens (NA 0.85, LCPlanFLN, OLYMPUS)

• Experiments other than the above. Digital biological microscope (GR-D8T2, Shodensha, Inc.) with the 40 × objective lens (NA 0.65, Shodensha, Inc.), or the 10 × objective lens (NA 0.25, Shodensha, Inc.)

Confirmation of stained cells

Confirmation of stained cells for each experiment was conducted as follows.

Figure 1 Principal component analysis (PCA) of absorption spectra of E.  gracilis in CM medium.

(A) First principal component (PC1) of absorption spectra of single cells (B) Eigenvalues of PC1 to PC3 in single cell absorption spectra and the numbers of cells for PCA (C) Bright field microscopic image of cells with the inverted microscope (IX71, OLYMPUS) with the 100× objective lens of NA 0.85 (LCPlanFLN, OLYMPUS) Details of each sample are summarized in Table 4. Since the BC solution is transparent, it does not itself stain cells (also for Figs. 2 to 5).

Figure 2 PCA of absorption spectra of E.  gracilis in CM medium mixed with MP.

(A) First principal component (PC1) of absorption spectra of single cells (“Pigment (CM)” and “Pigment (BC)” were raw spectra without PCA, scaled to fit the range of vertical axis.) (B) Eigenvalues of PC1 to PC3 in single cell absorption spectra and the numbers of cells for PCA (C) Bright field microscopic image of cells with the inverted microscope with the 100× objective lens (details of each sample are summarized in Table 4.)

Cells were observed by a bacteria counter (A-161, Sunlead Glass Co., Ltd., Saitama, Japan) and a bright field microscope (GR-D8T2) with the 10 × and 40 × objective lenses once a day for 2 days.

Figure 3 PCA of absorption spectra of E. gracilis in CM medium mixed with AP.

(A) First principal component (PC1) of absorption spectra of single cells (“Pigment (CM)” and “Pigment (BC)” were raw spectra without PCA, scaled to fit the range of vertical axis.) (B) Eigenvalues of PC1 to PC3 in single cell absorption spectra and the numbers of cells for PCA (C) Bright field microscopic image of cells with the inverted microscope with the 100× objective lens (details of each sample are summarized in Table 4.)

Figure 4 PCA of absorption spectra of E. gracilis in CM medium mixed with TB.

(A, B) First principal component (PC1) of absorption spectra of single cells (“Pigment (CM)” and “Pigment (BC)” were raw spectra without PCA, scaled to fit the range of vertical axis). (C) Eigenvalues of PC1 to PC3 in single cell absorption spectra and the numbers of cells for PCA. (D) Bright field microscopic image of cells with the inverted microscope with the 100× objective lens (details of each sample are summarized in Table 4.)

Figure 5 PCA of absorption spectra of E. gracilis in CM medium mixed with MB.

(A, B) First principal component (PC1) of absorption spectra of single cells (“Pigment (CM)” and “Pigment (BC)” were raw spectra without PCA, scaled to fit the range of vertical axis). (C) Eigenvalues of PC1 to PC3 in single cell absorption spectra and the numbers of cells for PCA (D) Bright field microscopic image of cells with the inverted microscope with the 100× objective lens (details of each sample are summarized in Table 4.)

Searching for an appropriate food pigments in viability assay for E. gracilis (Table 1)

Sample preparation for searching for an appropriate food pigments in viability assay.

Cells were cultured to a cell density 2.5 × 106 cells/mL for 36 days. This suspension of 0.5 mL was injected into a micro test tube containing 5 mg of food pigment (1% (W/V)) and thoroughly mixed by pipetting. Eight kinds of natural pigments were tested (Table 1).

Confirmation of stained cells with 8 food pigments.

Cells were observed twice, immediately after mixing the pigment and after 25 h.

Confirmation of reliable staining of dead cells (Tables 2 and 3)

Sample preparation for confirmation of reliable staining of dead cells.

Cells were cultured to a cell density 1.6 × 106 cells/mL for 21 days. The suspension was mixed with pigment. The composition of each sample is shown in Table 2.

Preparation of dead cells of E. gracilis.

Dead cells were obtained by the following two treatments (Table 3).

(1) Cells were treated with 0.2%(W/V) benzalkonium chloride (BC) solution (T1). 10%(W/V) BC solution (NIHON PHARMACEUTICAL CO., LTD) was used. Since the BC solution is transparent, it does not itself stain cells.

(2) Cell suspension was treated with microwave at 2.45 GHz until it boiled (T2).

Confirmation of stained cells and measurement of staining time.

The time from the mixing of pigment until all the cells in the observation area were stained was measured. The results are shown in Table 3.

Measurement of absorbance of single live or dead cells in the culture with food pigment or synthetic dye (Figs. 1–5)

Sample preparation for measurement of absorbance spectra of single cells.

Cells were cultured to a cell density 8.1 ×105 cells/mL for 9 days. It was divided into the following three kinds of experiments (Table 4).

• Staining of cells in normal culture environment (Naturally Alive/Dead) Cells were measured by absorbance imaging after incubation for 2 days with pigment.

• Staining of dead cells treated with microwave (MW Dead)

• After microwave treatment of the cell suspension, pigment was added and absorbance imaging was measured.

• Staining of dead cells treated with 0.2%(W/V) benzalkonium chloride (BC Dead)

After BC treatment of the cell suspension, pigment was added and absorbance imaging was measured. Since the BC solution is transparent, it does not itself stain cells.

On measurement of absorbance imaging, each sample was diluted 10-fold with CM medium and injected into a glass bottom dish (Matsunami glass D11130H).

As a control sample, pigment solutions were prepared with fresh CM medium as solvent (Table 4).

• Pigment (CM): pigment solution in which the solvent is CM medium

• Pigment (BC): pigment solution in which the solvent is CM medium with 0.2% BC

These control samples were injected into a glass bottom dish without dilution.

Then, samples were measured by scan-free, non-invasive absorbance spectral imaging A (x, y, λ) microscopy (Isono et al., 2015) (Details are described below).

Table 1 Staining ability of natural edible pigments for dead cells.

Cells were observed with a bacteria counter (A-161, Sunlead Glass Co., Ltd.) and a bright field microscope (GR-D8T2, Shodensha, Inc.) with the 10×objective lens (NA 0.25, Shodensha, Inc.) and with the 40×objective lens (NA 0.65, Shodensha, Inc.) 25 h after the pigment was mixed. All the pigments are derived from foods and purchased from Watashinodaidokoro Co., Ltd. The names of these pigments were translated into English from Japanese.

Color	Black	Brown	Purple	Blue	Green	Yellow	Pink	Red	
Food	Bamboo charcoal	Kaoliang	Purple sweet potato	Spirulina	Mixture of Yellow and blue gardenia	Yellow gardenia	Red beet	Monascus	
Staining	×	×	∘	×	×	×	×	∘	
Notes.

∘ Identifiable staining of dead cells.

× Unidentifiable staining of dead cells.

Table 2 Composition of samples for staining of dead cells.

Samplea	Pigment concentration %(W/V)	Pigment [mg]	TBb 1% (W/W) [µL]	MBc 1% (W/W) [µL]	Cell suspension (CM medium) [µL]	BCd 10% (W/V) [µL]	
CM	0	0	0	0	196	4	
MP	1	2	0	0	196	4	
AP	1	2	0	0	196	4	
TB (0.05%)	0.05	0.1	0	10	186	4	
TB (0.3%)	0.3	0.6	60	0	136	4	
MB (0.05%)	0.05	0.1	0	10	186	4	
MB (0.3%)	0.3	0.6	60	0	136	4	
Notes.

a CM, Cell Suspension (CM medium); MP, Monascus pigment; AP, Purple sweet potato; TB, Trypan blue (TOKYO CHYEMICAL INDUSTRY Co.,Ltd.); MB, Methylene blue (KOKUSAN CHEMICAL Co., Ltd.).

b The solvent of 1% (W/W) TB solution was purified water.

c The solvent of 1% (W/W) MB solution was purified water.

d BC, 10% (W/V) benzalkonium chloride solution (NIHON PHARMACEUTICAL CO., LTD). Since the BC solution is transparent, it does not itself stain cells.

Table 3 Results of staining of dead cell treated with microwave or BC.

Cells were observed with a bacteria counter (A-161) and a bright field microscope (GR-D8T2) with the 10 and the 40×objective lens after pigment was mixed (Table 2).

Pigment	T 1	T 2	
	Time [min]	Staining	Time [min]	Staining	
MP	3	∘	2.5	∘	
AP	1	∘	1	∘	
TB (0.05%)	∼0.5	△	3.5	∘	
TB (0.3%)	∼0.5	∘	3	∘	
MB (0.05%)	∼0.5	△	∼0.5	∘	
MB (0.3%)	∼0.5	△	∼0.5	∘	
Notes.

T1 The pigment was mixed after cells were treated with benzalkonium chloride solution.

T2 The dye was mixed after the cells were treated with microwaves.

Time min The time from the mixing of pigment until all the cells in the observation area were stained. The time of “∼0.5” means that staining was completed within 0.5 min.

∘ Identifiable staining of dead cells.

△ Cells were stained, however, because they were thin, it is difficult to distinguish them from living cells.

Table 4 Pigment concentrations and condition of samples for absorbance spectral imaging (Figs. 1 to 5).

	
Notes.

a Naturally Alive/Dead: Staining of cells in normal culture environment.

MW Dead, Dead cells treated with microwave.

BC Dead, Dead cells treated with benzalkonium chloride (The final concentration of benzalkonium chloride was adjusted to 0.2% by stock solution (10%).).

Pigment (CM), Pigment solution in which the solvent is CM medium.

Pigment (BC), Pigment solution in which the solvent is CM medium with 0.2% BC.

b Times from mixture of pigment with cell suspension to measurement.

c Dilution rate at the time of measurement of absorbance imaging by CM medium.

d Since the BC solution is transparent, it does not itself stain cells.

Measurement methods of single-cell absorbance.

Detailed methods of scan-free absorbance spectral imaging were previously described (Isono et al., 2015) and the subsequent improvement was reported recently (Yamashita et al., 2018; Yamashita et al., in press). Here we give only brief specifications.

The system for scan-free (snapshot) absorbance spectral imaging is composed of an inverted microscope (IX71, OLYMPUS), a custom-made fiber bundled array to convert a 2-dimensional image to the 1-dimensional slit image, an imaging spectrometer with a large imaging area of 27 × 27 mm2 (f = 32 cm, IsoPlaneSCT-Advance, Roper Scientific, Planegg, Germany), and an electrically-cooled CCD camera (2, 048 × 2, 048 pixels with 13.5 µm pixel size, PIXIS2048BUV, Roper Scientific). For the fiber bundled array (2D-1D converter), a 50-µm core/55-µm clad/60-µm coating silica fibers are assembled to 16 ×16 (0.96 × 0.96 mm2) array on the input side and to 1 ×256 (0.06 × 15.36 mm2) array on the output (slit) side. On the side port of the inverted microscope, the ×100 magnified image of the sample was focused on the 16 ×16 2D array (x × y), which is rearranged in order into the 1D array to fit the entrance slit of the spectrometer. The vertical size of both the imaging area (27 mm) and the CCD (27.65 mm) covers that (15.36 mm) of the 1D array of the 2D-1D converter, so that the whole 3D image A(x, y, λ), a datacube of 16 × 16 × 2, 048, is obtained simultaneously without need for the vertical scan. To be precise, the vertical length in the imaging area where an aberration-corrected image is obtained is limited by 14 mm, so that the images from 22–23 fibers at both ends of the 1D array are not aberration-free. The shortest acquisition time is 0.05 s which is limited by the response time of the mechanical shutter of the spectrometer.

The glass bottom dish with sample was set on the inverted microscope and observed with the 100 × objective lens of NA 0.85 (OLYMPUS, LCPlanFLN) from below. The light source was a 150 W Xenon lamp (Hamamatsu Photonics, Hamamatsu, Japan) to illuminate a region of 2 mm in diameter of the sample from above through a condenser. The intensity (photon flux density) on the sample was 1,800 to 2,100 µmol/m2/s for 0.5 s exposure. The transmitted light was transferred through the objective and a focusing lens to the side port of the microscope. The absorbance of the sample was calculated assuming that the transmitted light intensity in the region where only the culture solution exists is 100%. The spectrometer has automatically exchangeable three gratings, 1,200, 300, and 150 grooves/300 nm blaze. We used the 150 grooves grating with wavelength resolution of 1.1 nm at 546 nm (50 µm slit width, determined by the core size of the fiber) and wavelength span from 390 to 790 nm. All the measurements were performed at room temperature.

Table 5 Composition of samples for the growth curve and dead cell ratio in the culture mixed with pigment (Figs. 6, 7).

Sample	Pigment concentration% (W/V)	Pigment[mg]	Cell suspensionb(CM medium) [mL]	Glucose [mg]	
CM	0	0	1	0	
CM+G	0	0	1	12	
MP	1	10	1	0	
MP+G	1	10	1	12	
AP	1	10	1	0	
AP+G	1	10	1	12	
TB	0.3	3	1	0	
TB+G	0.3	3	1	12	
MB	0.05	0.5a	0.95	0	
MB+G	0.05	0.5a	0.95	12	
Notes.

a The 0.05 mL of 1% (W/W) methylene blue solution with CM medium as a solvent.

b “MB” and “MB+G” are 0.05 mL less than other samples. The difference in this amount can be ignored from the result of 5 times of cell density measurement of the following sample.

Sample 1, Cell suspension approximately equal to the initial cell density in this test 6.39 × 105 ± 2.2 × 104 cells/mL (Average ± Standard deviation).

Sample 2, 95% (V/V) of Sample1 diluted with purified water. 6.29 × 105 ± 2.3 × 104 cells/mL (Average ± Standard deviation).

Figure 6 Growth curve of E. gracilis in pigment mixed culture.

(A) Growth curve of E. gracilis in CM medium mixed with pigment (B) Growth curve of E. gracilis in CM medium mixed with pigment and glucose.

Figure 7 Ratio of dead cells in pigment mixed culture.

(A) Ratio of dead cells of E. gracilis in CM medium mixed with pigment (B) Ratio of dead cells of E. gracilis in CM medium mixed with pigment and glucose.

Principal component analysis (PCA) of absorption spectra.

Absorption spectra obtained from different cells in each sample were principal component analyzed. The spectra of the first principal component (PC1), the eigenvalues of PC1 to PC3, and the number of cells for PCA are shown in Figs. 1–5. For the PCA, web application was used (Easy, 2019).

Growth curve and ratio of dead cells of E. gracilis in culture mixed with pigment (Figs. 6 and 7)

Sample preparation for the growth curve and the ratio of dead cells.

Cells were cultured to a cell density 5.6 × 105 cells/mL for 5 days. The suspension was mixed with pigment. The composition of each sample is shown in Table 5. The concentration of glucose was adjusted to be equivalent to that in KH medium (Kitaoka, 1989) which is a common heterotrophic medium of E. gracilis.

Cell counting method for the growth curve (Fig. 6).

The count for the growth curve was performed using plankton counter plates (MPC-200, Matsunami Glass Ind.,Ltd.). Each sample was appropriately diluted with a solution of 10%(W/V) BC solution diluted 50-fold with purified water, and cells were fixed and counted. For the counting, a bright field microscope (GR-D8T2) with the 10 × and 40 × objective lenses were used. The counting was performed once a day for 5 days. The results are shown in Fig. 6.

Cell counting method for the ratio of dead cells in culture mixed with pigment (Fig. 7).

The same sample as above section was used, and these measurements were performed nearly at the same time. The cell counting was performed with a bacteria counter and a bright field microscope (GR-D8T2) with the 10 × and 40 × objective lenses. The ratio of stained (dead) cells to all the cells contained in the observation sections are shown in Fig. 7. For viability assay of pigment free samples “CM” and “CM+G”, those cells that lost motility and vivid green color in their chloroplasts were judged to be dead cells. However, since the resolution was low at a magnification with the 10 × objective lens for this judgment, they were observed with the 40 × objective lenses. The counting was performed once a day for 5 days. The total number of cells counted was over 50 cells at the start of the incubation in the culture mixed with the pigment (0 h), over 100 cells on the 2nd day (18 h), and over 200 cells from the 3rd day.

Appropriate concentration of natural pigments for viability assay (Tables 6 to 8)

We searched for concentrations of MP and AP (in CM medium at low pH (3.5)) that are easy to distinguish dead cells and maintain health of living cells. Samples of various concentrations of pigments were prepared (shown in Tables 6 and 7) and the state of staining and motility of cells were observed once a day. The observation was performed with a bacteria counter and a bright field microscope (GR-D8T2) with the 40 × objective lenses.

Table 6 Composition of samples for appropriate concentration of natural pigments for viable assay.

MP % (W/V)	0.3	0.5	0.6	0.7	0.8	0.9	1	1.5	2	3	
AP % (W/V)	0 (control)	0.05	0.1	0.2	0.3	0.5	1	1.5	2	3	
Notes.

Total volume of sample : 1 [mL] (Solvent is CM medium.).

Initial cell density : 4.0 × 105 [cells/mL].

(Cells were cultured for 4 days before pigment was mixed and observed for 6 days).

Table 7 Composition of samples for appropriate concentration of MP for viable assay.

MP % (W/V)	0 (control)	4	5	6	7	8	
Notes.

Total volume of sample : 1 mL (Solvent is CM medium.).

Initial cell density : 7.0 × 105 cell/mL.

(Cells were cultured for 7 days before pigment was mixed and observed for 4 days).

Table 8 Summary of MP and AP for viability assay of E. gracilis revealed.

Pigment	Color	Concentrationa	Time untilstainingb	Cytotoxicity	Light stability	Expiration date	
MP	Red	0.9∼1.5 % (W/V)	3 min∼	Low	Unstable	4(∼6) days c	
AP	Red purple	0.5∼1.0% (W/V)	3 min∼	High (without glucose) Low (with glucose)	Stable	6 days	
Notes.

a MP, Concentration at which staining of dead cells and health of living cells are compatible.

AP, Concentration at which dead cells are clearly stained and cell health is relatively unaffected.

b Time of staining of dead cells treated with BC or microwave.

c Fading appears after 5 days, but staining of dead cells can be visually confirmed up to 6 days.

Results and Discussions

Searching for an appropriate food pigments in viability assay for E. gracilis (Table 1)

Among the food pigments summarized in Table 1, with the pigments extracted from Monascus (MP) and purple sweet potato (AP), dead cells of E. gracilis were stained as vividly as to be clearly distinguished from live cells. With the other pigments, by contrast, dead cells were not stained clearly enough to be visually distinguished.

Confirmation of reliable staining of dead cells (Table 3)

Table 3 shows the results of staining of dead cells treated with benzalkonium chloride (BC) or microwave. It was confirmed that in all the samples, dead cells were reliably stained. However, in the samples with the synthetic dye mixed and BC treatment (TB (0.05%), MB (0.05%) and MB (0.3%) of T1), dead cells were stained instantly, but since the whole cells became so lightly blue that they were difficult to distinguish from the green color of the chloroplasts of living cells. In TB (0.3%) of T1, dead cells were stained clearly enough to distinguish them from live cells. In all the samples with the synthetic dyes mixed and microwave treatment, dead cells were clearly stained. This suggests that the binding of the synthetic dyes to dead cells was weakened by the presence of BC. TB clearly stained dead cells by increasing the concentration, but there was no noticeable change in MB. On the other hand, in all the samples of the natural pigments, all dead cells were clearly stained. This result suggests that they are less susceptible to BC than the synthetic pigments to have more general versatility.

Measurement of absorbance of single live or dead cells in the culture with natural food pigment or synthetic dye (Table 4, Figs. 1–5)

Absorption spectra of E. gracilis in CM medium (Fig. 1)

Figure 1A shows the first principal component (PC1) of absorption spectra of E. gracilis cultured in CM medium. The absorption spectra excluding “BC Dead” are dominated by that of the chloroplast, which has two absorption maxima attributed to Chl-a around 430 and 680 nm. The spectra of these cells are difficult to distinguish, since the characteristic of Chl-a, which is the main component of living cells, is remarkable. On the other hand, “BC Dead” had no characteristic spectral structure, indicating that Chl-a in the cells were destroyed by BC.

Since the eigenvalues of each sample are close to 1 as shown in Fig. 1B, the spectra of each sample can be represented by PC1. “MW Dead” has smaller eigenvalues than other samples. This indicates that the damage of Chl-a caused by microwave treatment has a relatively large variation. In the bright field microscopic observation (100 × objective lens, NA 0.85), it is possible to distinguish between “Naturally Alive” and other samples of dead cells (Fig. 1C), but it is difficult with low magnification (10 × objective lens, NA 0.25).

Absorption spectra of E. gracilis in CM medium mixed with MP (Fig. 2)

Figure 2A shows the PC1 of absorption spectra of E. gracilis cultured in CM medium mixed with MP. Pigment solutions without cells (“Pigment (CM)” and “Pigment (BC)”) are characterized by having two absorption maxima around 425 and 515 nm. Spectra of dead cells are all characterized by an increase in the absorbance towards shorter wavelengths in the range of 490 to 600 nm. This is thought to be due to the MP, indicating that the substances in the cell are bound to it. In “BC Dead”, because there is a Q peak (672 nm) of Chl-a, which is lost in “BC Dead” in Fig. 1A without MP, one can tell that MP suppresses destruction of Chl-a by BC. On the other hand, “Naturally alive” shows almost the same structure as that of Fig. 1.

Since the eigenvalues of each sample are approximately 1, the spectrum of each sample can be represented by PC1 (Fig. 2B). It shows that all the samples of dead cells were stained evenly by MP. In the microscopic observation (100 × objective lens), one can readily distinguish between “Naturally Alive” and other samples of dead cells (Fig. 2C), even at low magnification (10 × objective lens).

Absorption spectra of E. gracilis in CM medium mixed with AP (Fig. 3)

Figure 3A shows the PC1 of absorption spectra of E. gracilis cultured in CM medium mixed with AP. Pigment solutions without cells (“Pigment (CM)” and “Pigment (BC)”) are characterized by having an absorption maximum around 530 nm. It is known that AP extracted from purple sweet potato has absorption maximum at 530 nm (He et al., 2015). Spectra of dead cells are all characterized by an increase in the absorbance towards shorter wavelengths in the range of 540 to 645 nm. This is thought to be due to the AP, indicating that the substances in the cell are bound to it. In “BC Dead”, because there is a Q peak (672 nm) of Chl-a, one can tell that AP suppresses destruction of Chl-a by BC as well as MP. On the other hand, “Natural alive” shows almost the same structure as that of Fig. 1.

Since the eigenvalues of each sample are approximately 1, the spectrum of each sample can be represented by PC1 (Fig. 3B). It shows that all the samples of dead cells were stained evenly by AP. In the microscopic observation (100 × objective lens), one can readily distinguish between “Naturally Alive” and other samples of dead cells (Fig. 3C), even at low magnification (10 × objective lens).

Absorption spectra of E. gracilis in CM medium mixed with TB (Fig. 4)

Figures 4A, 4B shows the PC1 of absorption spectra of E. gracilis cultured in CM medium mixed with TB. Pigment solutions without cells (“Pigment (CM)” and “Pigment (BC)”) are characterized by having an absorption maximum around 585 nm. “Pigment (BC)” has another absorption maximum at 625 nm. The absorption spectra of dead cells have a characteristic broad absorption maximum in the range of 550 to 590 nm not present in living cells. This is thought to be due to the TB, indicating that the substances in the cell are bound to it. There are two kinds of staining colors of dead cells, blue and purple. The spectra of dead cells stained purple have a broadened absorption maximum around 680 nm of Chl-a, whereas the spectra of dead cells stained blue do not. Therefore, when TB coexists with Chl-a within the cells, dead cells are stained purple, and when Chl-a does not exist, they are stained blue. Since the absorption maxima around 420 nm are also found in dead samples having no Chl-a absorption maximum around 680 nm, it is considered to be due to other intracellular pigments. In addition, as blue stained cells increased with time, TB destroys Chl-a although it does not as rapidly as BC. On the other hand, “Naturally alive” shows almost the same structure as that of Fig. 1.

Since the eigenvalues of each sample are approximately 1, the spectrum of each sample can be represented by PC1 (Fig. 4C). It shows that all the samples of dead cells were stained evenly by TB. Note that the above result is obtained by dividing the PCA into the two kinds of colors both in “Naturally Dead” and “BC Dead”. In the microscopic observation (100 × objective lens), one can readily distinguish between “Naturally Alive” and other samples of dead cells (Fig. 4D), even at low magnification (10 × objective lens).

Absorption spectra of E. gracilis in CM medium mixed with MB (Fig. 5)

Figures 5A, 5B shows the PC1 of absorption spectra of E. gracilis cultured in CM medium mixed with TB. Pigment solutions without cells (“Pigment (CM)” and “Pigment (BC)”) are characterized by having two absorption maxima around 610 and 665 nm respectively. The absorption spectra of dead cells have absorption maxima near these two absorption maxima. This is thought to be due to the MB, indicating that the substances in the cell are bound to it. There are two kinds of staining colors of dead cells, blue and blue–green (Fig. 5D). Blue–green is shown as “Green” in the figure. Since one of the absorption maxima (665 nm) is close to the absorption maximum of Chl-a (680 nm), it is difficult to judge the presence of Chl-a, but the spectra of the cells stained green seems to contain more Chl-a due to the higher absorbance in the range of 390 to 500 nm than the blue stained cells. Therefore, when MB coexists with Chl-a, dead cells stained blue–green, and when Chl-a does not exist, they stained blue. In addition, as blue stained cells increased with time, MB destroys Chl-a although it does not as rapidly as BC. On the other hand, “Naturally alive” shows almost the same structure as that of Fig. 1.

Since the eigenvalues of each sample are close to 1, the spectra of each sample can be represented by PC1 (Fig. 5C). Note that the above result is obtained by dividing the PCA into the two kinds of colors both in “Naturally Dead” and “BC Dead”. In the microscopic observation (100 × objective lens), one can readily distinguish between “Naturally Alive” and blue stained cells (Fig. 5D), even at low magnification (10 × objective lens). However, it is difficult to visually distinguish between blue–green stained cells and living cells.

Growth curve and ratio of dead cells of E. gracilis in CM medium mixed with pigment (Figs. 6 and 7)

Growth curve of E. gracilis in CM medium mixed with pigment (Fig. 6A)

Figure 6 shows the growth curve of E. gracilis in CM medium mixed with pigment. In the glucose-free culture with pigment contained, the cell density for the natural pigment (AP, MP) increased almost equally to that for the control. However, there was no significant increase for the AP after 65 h. On the other hand, in the culture containing the synthetic dyes (TB, MB), the cell density was always kept at a lower value than for the control with no significant increase observed during the incubation period. Therefore, the natural pigment does not inhibit growth of E. gracilis for three or four days, but the synthetic dye inhibits it (Fig. 6A).

Growth curve of E. gracilis in CM medium mixed with pigment and glucose (Fig. 6B)

In the culture containing pigment and glucose, the cell density for the natural pigment (“AP+G”, “MP+G”) was almost equal to or higher than that for the control. Especially after 90 h the cell density was larger than for the control to indicate a larger increase rate. On the other hand, in the culture containing the synthetic dye (“TB+G”, “MB+G”), the cell density remained lower than for the control during the incubation period. However, for “MB+G”, a significant increase of the cell density was observed compared to MB. Therefore, it was found that addition of glucose to the culture mixed with pigment improves the cell viability. A large improvement in the cell viability, particularly in AP and MB due to the addition of glucose, suggests that these pigments have a negative effect on the metabolic system of glucose (Fig. 6B).

Ratio of dead cells of E. gracilis in CM medium mixed with pigment (Fig. 7A)

In the glucose-free culture containing pigment, the ratio of dead cells for MP stably maintained to be 10% or less during the incubation period. By contrast, the ratios for AP and TB increased with time. The ratio for MB was comparable to the ratio of MP until 60 h, but it increased sharply after that.

Ratio of dead cells of E. gracilis in CM medium mixed with pigment and glucose (Fig. 7B)

In the culture containing pigment and glucose, the ratio of dead cells of pigments for other than “TB+G” did not show a significant increase. In particular, the increase of the ratio for “AP+G” and that for “MB+G” after 60 h were largely suppressed. The ratio for “MP+G” was stably kept lower than that for MP.

MP is known to have an anti-bacterial activity, and its targets are diverse including gram-positive and gram-negative bacteria, yeast, filamentous fungi (Martínková, Jzlová & Veselý, 1995; Feng, Shao & Chen, 2012). As shown in Fig. 7A, however, it is most likely that E. gracilis is not affected by anti-bacterial activity by MP. On the other hand, when AP was added, dead cells increased over time as shown in Fig. 7A. AP exhibit anti-microbial activity through several mechanisms such as cell damage induced by destroying cell walls, membranes, and intercellular matrix, but its mechanism of the activity is not known in detail. In addition, it is known that anthocyanin suppresses gram-negative bacteria but not gram-positive bacteria (Khoo et al., 2017). It has been reported that AP derived from purple sweet potato inhibits the growth of Salmonella Enteritidis (gram-negative bacteria) (Cevallos-Casals & Cisneros-Zevallos, 2002). The present research is the first report that the anti-microbial test of MP and AP was carried out on microalgae such as E. gracilis because most of the test has been done on pathogenic bacteria. It is also suggested that AP suppresses the synthesis of glucose in cells because addition of glucose to CM medium to which AP is added can suppress the increase of dead cells of E. gracilis (Fig. 7). Furthermore, the living cells in the AP mixed CM medium showed the same motility and bright green color as the living cells in the CM medium, and the number of dead cells did not increase with the AP concentration. Therefore, it is considered that AP acts as a target on both dead cells and living cells susceptible to AP. Since similar results were obtained in experiments using violet cabbage powder (KENIS, Ltd., Osaka, Japan) as a natural pigment containing AP, it is most likely that AP has the greatest anti-microbial activity among the components contained in the edible pigment powder. The influence of AP on the metabolism of E. gracilis is a future subject to be elucidated.

Appropriate concentration of natural pigments for viability assay (Tables 6 to 8)

As a result of examining an appropriate concentration at which staining and health of cells was maintained over a long period, the concentration of MP was found to be 1.5%(W/V). With this concentration, it was enabled to identify stained dead cells for 6 days while maintaining the health of living cells. However, fading of staining was observed after 4 days. More fading was observed at 0.9 to 1%(W/V), but it was enabled to identify stained dead cells for 6 days. At 2%(W/V) or more, although the number of swimming cells decreased, cells that did not swim were alive because they moved flagella or showed stretching motions, and the ratio of stained cells on the 4th day was roughly 10%.

Because AP tended to increase stained cells in proportion to the concentration, it is difficult to balance the staining of the cells with the health of them, but the balance is relatively good between 0.5 and 1%(W/V). However, even at 0.05%(W/V), since dead cells increased compared with the control sample, it is necessary to use it together with glucose for long-term culture. During the observation period (6 days), since AP contained in dead cells and in the culture did not fade, the stability against light was found to be higher than MP. This is probably due to the fact that AP is a stable pigment under acidic conditions (Khoo et al., 2017). Therefore, it is convenient to use it in combination with CM medium which is acidic (pH 3.5). In addition, from Fig. 3, it is confirmed that the absorption maximum of AP in the solution (not bound to cells) is 529 nm, but its maximum value largely shifts to 540 nm by binding with substances in the cell.

From the above results, it is important to select the pigment properly depending on the purpose and target cells. For example, it may be one of the future tasks to combine MP and AP and to search for the optimum blending ratio for staining dead cells in a shorter time and for longer-term monitoring of cell viability. In addition, since there are many variations of natural pigments, their application range is wide. In particular, it can be applied to cell counting by flow cytometry, since strong staining of dead cells is obtained by selecting an appropriate pigment. It may also contribute to elucidation of the metabolic pathway by interaction between a pigment and a living cell.

Conclusion

We examined the feasibility of eight types of edible natural pigments to the DET for green algae and found that MP and AP have high ability to determine viability of E. gracilis. Blue, green, and yellow natural pigments did not stain a deep green cell E. gracilis so strongly as to be visually discriminated.

As well as the conventional synthetic viability assay reagents TB and MB, the natural pigments MP and AP stained visually with dead cells of E. gracilis. Further, the natural pigments showed more clear staining of dead cells than the synthetic dyes in the solution containing BC.

From the absorption spectra by PCA, it was found that the natural pigments stained dead cells in one color, whereas the synthetic dyes stained dead cells into two colors depending on Chl-a content. Especially in MB, it is impossible to distinguish these cells (Fig. 5D, “Naturally Alive”, “Naturally Dead (Green)”, “BC Dead (Green)”) by only the color of the images with microscopic observation. On the other hand, comparing the absorption spectra of these cells by absorbance spectral imaging, MB spectrum can be confirmed only in dead cells. In natural pigments, it was confirmed that such cells did not exist (Figs. 2 and 3C).

Also, since the natural pigments did not positively destroy Chl-a, a variation of the obtained spectra were small, whereas the synthetic dyes destroyed Chl-a, resulting in a larger variation of the spectra obtained. In addition, it was confirmed that the natural pigments do not show marked growth inhibition or toxicity to living cells as compared with the synthetic dyes. In fact, samples with the natural pigment showed a higher cell density than the control sample in CM medium. For this performance (higher growth rate than the control), however, AP requires addition of glucose. It was confirmed that the pigments bound to the dead cells can be monitored for the cell viability for 6 days. In addition, their cost is one tenth of that of TB currently used widely as a reagent for DET, and they are environmentally friendly.

MP can stain dead cells in a short time, but because it is unstable to light, it is not suitable for long-term monitoring of the cell viability. On the other hand, AP is suitable for long-term monitoring of the cell viability because it is stable to light, although AP needs addition of glucose for survival of cells. Therefore, it is important to grasp the characteristics and the influence of each pigment on the target cell, and it is a future task to search for the optimum use condition depending on each purpose. In food, hygiene, and life science field, they can play an important role from special facilities like laboratories and factories in everyday environments where more safety is required such as cafeterias and families.

The following patent on this manuscript is pending.

Title: Method and kit for cell viability assay

Patent Application Number: 2018-241789

Date: Dec. 25, 2018

Supplemental Information

Supplemental Information 1 Raw data

Click here for additional data file.

Additional Information and Declarations

Competing Interests

Author Contributions

Patent Disclosures

Data Availability

Koji Yamada and Kengo Suzuki are employed by Euglena Co., Ltd. The patent (Patent Application Number: 2018-241789) on this research is pending.

Kyohei Yamashita conceived and designed the experiments, performed the experiments, analyzed the data, contributed reagents/materials/analysis tools, prepared figures and/or tables, authored or reviewed drafts of the paper, approved the final draft.

Koji Yamada and Kengo Suzuki contributed reagents/materials/analysis tools.

Eiji Tokunaga contributed reagents/materials/analysis tools, authored or reviewed drafts of the paper, approved the final draft.

The following patent dependencies were disclosed by the authors:

Patent Application Number: 2018-241789

Title : (In Japanese) 細 胞 の 生 死 判 別 方 法 及 び 細 胞 の 生 死 判 別 用 キ ッ ト

Title: (English translation) Method and kit for cell viability assay

Date: Dec. 25, 2018

Inventors: Kyohei Yamashita and Eiji Tokunaga

Patent applicant: Tokyo University of Science

The following information was supplied regarding data availability:

The raw data are available in the Supplemental File.

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
