# Peer review of "Noninvasive and safe cell viability assay for Euglena gracilis using natural food pigment"

_PeerJ, doi:10.7717/peerj.6636_

## Round 0.1 · original submission · Major Revisions

Two reviewers acknowlege values of your paper. However, they also recognize many problems to be solved before publication as listed in the reviewer's comments. Please revise your manuscript according to their suggestions.

In addition to the reviewers comments, I find additional issues to be solved. Please see below.

1) Figures 1-3 shows the data from a limited number of cells. For example, the spectrum for "Living" specimen is shown based on the data obtained from one cell. Please show that the data is representative of cell population or statistically correct. At least one more or two more cell data should be added.
2) In the Figure legend, the magnification of the microsocpic images are indicated. But this value is sometimes meaningless because it depends on the printing size. Therefore you can remove these statements. Instead, information of the microscope and the objective lens (e.g. Olympus ULWD CDPlan x20 Ph, NA=xx,,,) used for these experiments should be added in the materials and methods section or figure legends.

Reviewer 1 ·

Basic reporting

This is an interesting manuscript. However:
- The language would benefit from revision to improve clarity. Sentences are often complex and difficult to understand. E.g. lines 73-75 state that E. gracilis can grow in heterotrophic cultures by photosynthesis, which is probably not what the authors meant. There are too many examples of unclear text to define them all.
- The authors should use past tense to describe results and also results reported in the literature. Past and present tense should be used consistently throughout the manuscript. E.g. line 137 - "8 natural food pigments WERE tested in this study..." not currently being tested ("are") as indicated in the text.
- The authors should avoid use of "etc." and "and the like". If there are more relevant points which they wish to raise, please state them. If not, please don't leave the reader guessing at what is being left out.

- The introduction shows appropriate context and the literature is relevant, although the authors rely heavily on citing their own work, which may raise questions concerning the thoroughness of the background review. They could consider including methylene blue in their description of popular vital stains. Methylene blue is considered less toxic than trypan blue and in my experience is much more popular as well.
- The statemen in lines 107-109 "The mechanism for TB to stain cells ..." should be supported by citing relevant literature.

- The structure conforms PeerJ standards.

- The figures are relevant and of reasonable quality. However, both spectra and photographs would benefit from having clear labels. i.e. the peaks in the spectra should be clearly labelled and arrows used to indicate stained and unstained cells in the photos in figures 4, 5 and 6.
- Figures should be referred to where relevant. In general, Figures are not relevant to the section headings in the methods and should only be referred to in the methods if the figure illustrates the method - which I did not find to be the case in this manuscript.
- It was not possible to access the supplementary (raw) data files, which were not in a format accessible to me. The file extension did not appear to be one accepted by PeerJ either.

- The results are self-contained and relevant to the hypothesis, but quantitative data would be desirable.

Experimental design

- The research presented was original and with the journal scope.

- The research question (can non-toxic food dyes be used to stain algae?) was defined, but the description of why they considered it relevant and meaningful was not convincing. The manuscript would benefit if data would be provided which would demonstrate how the authors envision the stains being used in practice, along with control data obtained using currently available dyes. The authors claim that non-toxic stains are needed so that cells can be monitored in situ, but then always remove cells from the system in order to study them. They claim that cells should be monitored in situ so that the same cell can be monitored over time, but provide no suggestions as to how that same cell would be found within the larger population, if it were not removed for staining and was allowed to grow freely with its neighbours.

- The investigation does not appear to have been rigorously performed, although the ethical standard is okay. There is no quantitative data presented, and wherever numerical data is mentioned, it is only as an estimate. This diminishes the usefulness of the research and reduces the probability of the dyes being adopted for use outside the lab which has generated the results.

- The methods are difficult to understand and would be difficult to replicate. Improving the English will help improve method clarity. In addition, the authors should consider describing methods, rather than experiments. E.g. How were the the cells cultivated? What was the medium (-a) used? If C. reinhardtii was cultivated differently from E. gracilis, this should be in the cultivation section, not in a separate section where someone looking for methods might not find it. Did bright field microscopy differ depending on which samples were viewed? If so, this could be described in a microscopy section, not each variation being given its own section.

Validity of the findings

The paper presents spectra and photos obtained from staining the photosynthetic protest Euglena gracilis with 2 food-grade dyes. The authors demonstrate that the dyes penetrate and stain dead or dying cells, and speculate that this could be useful in monitoring the population over time. The work is interesting because good vitality stains are useful in many microbiological application. However, the paper is disappointing in that it does not include any quantitative data, nor any evidence of how they propose the stain(s) could be used in practice that would differ from the use of any currently used vital / viable stain.

- The results are of interest to the scientific community.

- No statistical analysis is provided. No quantitative data is provided. There is no evidence provided that staining the cells provided more accurate or useful data than assessing the population state without staining (see Figure 1 which clearly shows that dead cells of Euglena could be monitored without staining by making measurements arond 510 or 690 nm. Where is the experimental data comparing the results with and without staining? Where is the experimental data comparing the result with the new stains with a selection of old stains?

- Speculation is extensive in the conclusions. It goes well beyond the data provided in the results. That it is speculation is obvious to a knowledgeable reader, but it is not always clearly identified.
- The conclusions go beyond the original research question and are not supported by the results. E.g. The authors claim that a non-toxic stain is needed to allow in situ staining and monitoring, provide evidence that one of their stains is toxic to the organism being stained, and then conclude that this is a good non-toxic stain and recommend its continued use.

- The statement in lines 466-467 that "... this method... can be applied to the research such as basic study on the metabolic mechanism of the cell and breeding." has no basis in the results presented, nor do the authors indicate any way in which the method could be applied for this purpose.
- Claims concerning animal cell culture are irrelevant to the research presented.

Reviewer 2 ·

Basic reporting

The manuscript deals with viability test of microalgae Euglena gracilis with edible dyes. The reviewer suggests the author get some professional English proofreading, especially in table and figure legend section. (e.g. I could not understand the meaning of the legends of table 1 and 2) In some of parts, more references are needed. I would like to show detail in "General comments to authors" section. In my reviewing, I could not open .opj file with Origin viewer because of the permission trouble. So I could not say detail about the raw data.

Experimental design

In my understanding, the purpose of the experiment of the article was whether edible dye can used to determine the cellular viability of Euglena. Because of the shortage of the control experiment, their results could not show these facts.

Validity of the findings

The authors need to add the appropriate control experiments to improve the manuscript.

Additional comments

Line 134-136. They mentioned the possibility of natural pigment in adding antioxidant function. Synthetic food dyes also have the same possibility. The authors need to show the references.


Line 159. The author shows AP interacts with both cellulose and pectin. In my knowledge, Euglena does not contain significant amount of these polysaccharides. If you have further information about polysaccharide content of Euglena (whether they contains cellulose and pectin), please show references.

Line 166-167. The author mentioned the possibility to use the DET method with edible dye in animal cells if they succeed in the experiments using Euglena. Cell structures of Euglena and animal cells are completely different. So I think they cannot estimate the possibility whether they can use the method in the animal cells or not.


In my simple view, when I compare the spectrum in Figure 1, 2, and 3, Figure 1 spectrum shows the most significant difference between living cells and dead cells. The author need to show importance of DET process to determine cellular viability in the method.
The author did not set the positive control experiment whether the all dead cell stained with MP or AP dye. For example, when Euglena mixed with Lugol’s iodine solution (contains iodine and potassium iodide), all the cells dies immediately. (In the comment about Table 1 (e.g. line 237-238), the author suggest the states of staining of dead cells, but the reviewer could not understand how they determine these states.) They need further description especially in the method section (line 233-238).
I also think the author need to show the comparison between the DET results with edible dyes and traditionally used dye in Euglena viability assay e.g. trypane blue or propidium iodide, because the reviewer could not tell the results shown here is reasonable or not.

In the results of Figure 5, they mentioned stained cell rate were different between AP and MP. In my understanding, if the viability of Chlamydomonas can evaluate with DET method with AP and MP, the ratio with these dyes shows almost same value. How do you understand the result?

The meaning of the experiment around Figure 6 was difficult for me. Would you tell me why you did the experiment? What is the purpose of this experiment?

There are many small mistakes in English (e.g. spelling of “Monascus” in line 274). Please check whole manuscript again.

---

## Round 0.2 · Minor Revisions

One of the reviewers requires a numbers of revisions as listed below. I agree with this reviewer's opinions. I hope you find the suggestions useful to improve your paper. The reviewer requires mostly rewriting of the paper. This is why I decided as minor revisions. However, the reviewer also requires a couple experiments. If you need a time (more than 40 days) to complete these experiments, please let me know. You can taka a time if needed.

Reviewer 2 ·

Basic reporting

In previous review, I suggested some English proofreading, especially in table and figure legend section. But I think the author did not changed the legends completely. I could not understand the meaning of legend for Table 1 (I could not understand the first sentence of Table 1, especially. The expression "food of pigment" is difficult to understand the meaning what the authors want to say.), as I mentioned. In addition, Table 1 contains the author's experimental data. I recommend revision of the title. Title of Table 8 is also need to revise. The authors need to check titles of text section (especially in Materials and Methods section 4 and 5). I recommend the authors to get English proofreading by native English speaker.

Experimental design

In Abstract lines 16-18, the authors mentioned natural pigments showed growth promoting effect and synthetic dye showed inhibiting effect. There are no statistic information in Figures 6 and 7, so I could not tell difference between control and natral pigment groups (it looks almost same). If they would like to show difference between them, they need to show statistic information for Figures 6 and 7. The y-axis of Figure 6 shold be started 0 (not 4x10E5). The author would better to check the cell density range of Figure 7 (a) and (b). I could not understand the meaning of values in Table 5 (data for Figures 6 and 7). If the experiment were done as Table 5, cell suspension only added in CM, MB, and MB+G culture, and the volume of cell suspension in CM and other two were different. The growth curve experiment should be done in same cell concentration condition among all groups. And I think the data in Table 5 is something wrong. Method about Figure 6 (line 302), the author wrote "precipitated", and I could not understand the meaning of this process. The authour would like to say "centrifused to concentrate"? (In usual case, appropriate dilution is recommended. In the method, first dilute the sample, and after that concentrate it, I felt it is not very good process.)

The following is additional comment. The growth of both culture almost same (Fig.6, 2.5x10E6/ml in CM 92h vs 3.1x10E6/ml in CM+Glc 86h), but in usual case, growth of Euglena gracilis in glucose added medium showed much faster than autotrphic culture (around 10 times faster). I could not understand why CM+Glc did not increase the growth.

Validity of the findings

The fact that natural food dye could stain dead Euglena gracilis was clearly showed. This is the most strong point in the manuscript, and I agree. But, I could not agree their conclusions written in last sentence of Abstract and last sentence of Introduction. It looks overestimate, because cellular structure and component of Euglena gracilis is completely different from animal cells and bacterial cells used in the medical and industrial use, as I said in previous revision comments.

Additional comments

As comments in above section, use of natural food dye in determination of cellular viability is important finding. In contrast, I could not catch the importance of the usage of scan-free absorbance spectral imaging to find the effectiveness of natural food dye (because the staining of AP and MP were very clear in visible image with normal microscopy). I recommend to the author to emphasis the importance of scan-free absorbance spectral imaging system to find the fact in this paper.

---

## Round 0.3 · accepted · Accept

Thank you very much for sending the valuable paper. I am happy to inform you that your paper is now accepted for publication.

# Reviewer 2 ·

Basic reporting

no comment

Experimental design

no comment

Validity of the findings

no comment